# The Mobilome-Enriched Genome of the Competence-Deficient *Streptococcus pneumoniae* BM6001, the Original Host of Integrative Conjugative Element Tn*5253*, Is Phylogenetically Distinct from Historical Pneumococcal Genomes

**DOI:** 10.3390/microorganisms11071646

**Published:** 2023-06-23

**Authors:** Lorenzo Colombini, Anna Maria Cuppone, Mariana Tirziu, Elisa Lazzeri, Gianni Pozzi, Francesco Santoro, Francesco Iannelli

**Affiliations:** Laboratory of Molecular Microbiology and Biotechnology (LAMMB), Department of Medical Biotechnologies, University of Siena, Policlinico Le Scotte, V Lotto I Piano, Viale Bracci, 53100 Siena, Italy; lorenzo.colombini2@unisi.it (L.C.);

**Keywords:** *S. pneumoniae*, genome, mobilome, mobile genetic elements (MGEs), integrative conjugative elements (ICEs), integrative mobilizable elements (IMEs), genomic islands (GIs), prophages, transformation, competence

## Abstract

*Streptococcus pneumoniae* is an important human pathogen causing both mild and severe diseases. In this work, we determined the complete genome sequence of the *S. pneumoniae* clinical isolate BM6001, which is the original host of the ICE Tn*5253*. The BM6001 genome is organized in one circular chromosome of 2,293,748 base pairs (bp) in length, with an average GC content of 39.54%; the genome harbors a type 19F capsule locus, two tandem copies of *pspC*, the *comC1-comD1* alleles and the type I restriction modification system *SpnIII*. The BM6001 mobilome accounts for 15.54% (356,521 bp) of the whole genome and includes (i) the ICE Tn*5253* composite; (ii) the novel IME Tn*7089*; (iii) the novel transposon Tn*7090*; (iv) 3 prophages and 2 satellite prophages; (v) 5 genomic islands (GIs); (vi) 72 insertion sequences (ISs); (vii) 69 RUPs; (viii) 153 BOX elements; and (ix) 31 SPRITEs. All MGEs, except for the GIs, produce excised circular forms and *att*B site restoration. Tn*7089* is 9089 bp long and contains 11 ORFs, of which 6 were annotated and code for three functions: integration/excision, mobilization and adaptation. Tn*7090* is 9053 bp in size, flanked by two copies of IS*Spn7*, and contains seven ORFs organized as a single transcriptional unit, with genes encoding for proteins likely involved in the uptake and binding of Mg^2+^ cations in the adhesion to host cells and intracellular survival. BM6001 GIs, except for GI-BM6001.4, are variants of the pneumococcal TIGR4 RD5 region of diversity, pathogenicity island PPI1, R6 Cluster 4 and PTS island. Overall, prophages and satellite prophages contain genes predicted to encode proteins involved in DNA replication and lysogeny, in addition to genes encoding phage structural proteins and lytic enzymes carried only by prophages. ΦBM6001.3 has a mosaic structure that shares sequences with prophages IPP69 and MM1 and disrupts the competent *comGC*/*cglC* gene after chromosomal integration. Treatment with mitomycin C results in a 10-fold increase in the frequency of ΦBM6001.3 excised forms and *comGC*/*cglC* coding sequence restoration but does not restore competence for genetic transformation. In addition, phylogenetic analysis showed that BM6001 clusters in a small lineage with five other historical strains, but it is distantly related to the lineage due to its unique mobilome, suggesting that BM6001 has progressively accumulated many MGEs while losing competence for genetic transformation.

## 1. Introduction

Comparative genomics of *Streptococcus pneumoniae* has shown that 558 out of the average 1954 protein coding sequences are conserved [1]. Intraspecies variability is, in part, associated with allelic variants of genes encoding surface-exposed structures such as capsule, *pspC* and *pspA* genes [2,3,4,5,6]. In addition to these variable chromosomal loci, the majority of *S. pneumoniae* intraspecies variability is associated with the presence of mobile genetic elements (MGEs). The entire set of MGEs, collectively referred as a mobilome, contributes to the spread of antimicrobial resistance and virulence genes [7]. The comparison of invasive and noninvasive pneumococcal isolates shows that the invasive potential of a strain is often correlated to the presence or absence of a few specific additional sequences containing important virulence factors [8]. The pneumococcal mobilome includes prophages; plasmids; integrative conjugative elements (ICEs); pathogenicity islands; insertion sequences (ISs); and three families of small interspersed repeats, namely BOX elements, RUPs (repeat units of pneumococcus) and SPRITEs (*S. pneumoniae* rho-independent terminator-like element) [6]. Pneumococcal prophages are largely present in the genomes of *S. pneumoniae* isolates of different serotypes and are clustered into three major groups [9,10]. Satellite prophages are also widespread in *S. pneumoniae*, and despite being defective, may contain genes contributing to phenotypic traits such as virulence [11]. Plasmids are rarely found in *S. pneumoniae* genomes [6]. Most of the pneumococcal plasmids are identical or very similar to pDP1, a 3161 bp cryptic plasmid first identified in the type 2 D39 Avery’s strain and its derivatives [12,13]. ICE Tn*5253* was one of the first conjugative transposons identified in *S. pneumoniae* [14,15]. Tn*5253* is a 64.5 kb composite element containing Tn*5251*, another ICE carrying the tetracycline resistance *tet*(M) gene, and the Ω*cat*(pC194) element, carrying the chloramphenicol resistance *cat* gene [16]. Tn*5253* is able to (i) excise from the pneumococcal chromosome reconstituting the *att*B target site; (ii) produce circular forms; and (iii) transfer horizontally via conjugation in different bacterial species [17,18]. The chromosomal integration of Tn*5253* occurs downstream of a conserved 11 bp sequence of the essential *rbgA* gene in *S. pneumoniae* and in all other known hosts [19]. Pneumococcal pathogenicity island 1 (PPI1) is a highly variable accessory region widely present in *S. pneumoniae* isolates [20,21]. It is composed of (i) an operon coding for an ABC iron transporter, (ii) the *pezAT* toxin–antitoxin system, (iii) a putative lantibiotic production locus and (iv) the *nplT* neopullulanase gene. In particular, highly virulent *S. pneumoniae* isolates carrying a variant of PPI1 containing *pezAT* and *nplT* exhibit enhanced fitness in blood, lungs and nasopharinx [22]. In this work, we determined the complete genome sequence of the *S. pneumoniae* type 19F clinical isolate BM6001, which is the original host of ICE Tn*5253*. Sequence analysis allowed the definition of the BM6001 mobilome. Since prophage ΦBM6001.3 disrupts the competent *comGC*/*cglC* gene, we investigated whether prophage mitomycin C induction could restore competence for genetic transformation. Furthermore, a phylogenetic analysis of the BM6001 genome was performed to investigate whether the lack of competence for genetic transformation drives the accumulation of multiple MGEs.

## 2. Materials and Methods

### 2.1. Bacterial Strain and Growth Conditions

*S. pneumoniae* BM6001 is a serotype 19F clinical strain isolated from a patient with sinusitis [23]. BM6001 was grown in Tryptic Soy Broth (TSB) (BD) or in TSB supplemented with 1.5% agar (BD) and 3% defibrinated horse blood (Liofilchem, Roseto Degli Abruzzi, Italy) at 37 °C in the presence of a 5% CO_2_-enriched atmosphere.

### 2.2. Genomic DNA Purification

Bacterial cells were grown at 37 °C in 500 mL of TSB broth until reaching the late exponential phase (OD_590_ = 1), and then harvested via centrifugation at 5000× *g* for 30 min at 4 °C. High-molecular-weight genomic DNA was purified using a cetyl trimethyl ammonium bromide (CTAB)-based method [24]. Briefly, a cell pellet was dry vortex-mixed and lysed for 15 min at 37 °C in a lysis solution containing sodium dodecyl sulphate (SDS) 0.008% and sodium deoxycholate (DOC) 0.1%, as described in [25]. Proteins and polysaccharides were precipitated in 0.5 M NaCl and CTAB/NaCl (10% CTAB, 0.7 M NaCl) at 65 °C for 10 min. High-molecular-weight DNA was purified three times with 1 volume of chloroform/isoamyl alcohol (24:1 (*v*:*v*)), precipitated in 0.6 volumes of ice-cold isopropanol and spooled on a glass rod. DNA was resuspended in 10-fold diluted saline-sodium citrate (SSC) 1× buffer, adjusted to 1× SSC, homogenized using a rotator mixer and stored at 4 °C. DNA was quantified with a Qubit 3.0 Fluorometer (Invitrogen, Waltham, MA, USA) using the Qubit dsDNA BR Assay Kit (Thermo Fisher Scientific, Waltham, MA, USA) and with a spectrophotometer measurement (Implen, Munich, Germany). DNA integrity and size were assayed via horizontal gel electrophoresis using 0.6% Seakem LE (Lonza, Rockland, ME, USA) agarose in 0.5× Tris Borate EDTA running buffer.

### 2.3. Illumina Sequencing

Illumina sequencing was performed at MicrobesNG (University of Birmingham, Birmingham, UK) using a Nextera library preparation kit (Illumina Inc., San Diego, CA, USA) followed by MiSeq sequencing (Illumina Inc., San Diego, CA, USA) (2 × 250 bp paired-end sequencing). Illumina reads were trimmed using Trimmomatic v0.30 (https://github.com/usadellab/Trimmomatic) and analyzed with FastQC v0.11.5 (https://www.bioinformatics.babraham.ac.uk/projects/fastqc/).

### 2.4. Nanopore Sequencing

Nanopore sequencing was carried out essentially as already described [24,26]. Briefly, sequencing libraries were prepared in 1.5 mL LoBind tubes (Sarstedt, Nümbrecht, Germany) using wide bore (Φ 1.2 mm) tips for DNA manipulation to reduce physical shearing. The DNA size selection of the genomic DNA was obtained with 0.5 volumes of AMPure XP beads (Beckman Coulter, Milano, Italy) according to the manufacturer’s instructions. Then, 2.5 µg of size-selected DNA was employed for library construction by using the SQK-LSK 108 kit (Oxford Nanopore Technologies, Oxford, UK). Library preparation was performed according to the manufacturer’s protocol with the following modifications: (i) incubation on rotator mix for 15 min; (ii) the Library Loading Beads (LLB) were not added. Finally, 1 µg of DNA library was loaded onto a R9.4 flow cell (FLO-MIN106) (Oxford Nanopore Technologies). A 25 h sequencing run was performed on a GridION device (Oxford Nanopore Technologies). Real-time base-calling was performed with Guppy v3.2.6 (Oxford Nanopore Technologies), filtering out reads with a quality cut-off of <Q7. Base-called reads were analyzed with NanoPlot v1.18.2 (https://github.com/wdecoster/NanoPlot).

### 2.5. Genome Assembly and Annotation

BM6001 Nanopore reads were filtered to obtain a 100x coverage, taking 2 Mbp as the genome size estimate by using Filtlong v0.2.0 software (https://github.com/rrwick/Filtlong) with parameter *--target_bases* and assembled using Flye v2.7.1 (https://github.com/fenderglass/Flye). The resulting circular contig was polished with Medaka v0.7.1 (https://github.com/Nanoporetech/medaka) using the Nanopore reads, followed by two polishing rounds with Pilon v1.22 (https://github.com/broadinstitute/pilon) using the Illumina reads. Assembly completeness was assessed with Bandage v.0.8.1, whereas assembly quality was evaluated with both Ideel v5.5.4 (https://github.com/mw55309/ideel) and CheckM v1.1.3 (https://github.com/Ecogenomics/CheckM). Bwa v0.7.17 (https://github.com/lh3/bwa) and minimap2 v2.13 (https://github.com/lh3/minimap2) were used to align Illumina reads and Nanopore reads to the assembled genome, respectively. Read genome mapping was visualized with Tablet v1.17.08.17 (https://github.com/cropgeeks/tablet) and used to further verify the assembled structure. The BM6001 genome was automatically annotated with the NCBI Prokaryotic Genome Annotation Pipeline (PGAP) v5.1. Default parameters were used for all software unless otherwise specified.

### 2.6. Mobilome Analysis

The presence of ICEs and IMEs in the BM6001 genome was investigated with ICEfinder (https://bioinfo-mml.sjtu.edu.cn/ICEfinder/ICEfinder.html (accessed on 2 May 2023)), the presence of insertion sequences with ISsaga (http://issaga.biotoul.fr/issaga_index.php (accessed on 2 May 2023)), the presence of prophages with PHASTER (http://phaster.ca (accessed on 2 May 2023)) and the presence of genomic islands with IslandViewer4 (https://www.pathogenomics.sfu.ca/islandviewer/ (accessed on 2 May 2023)). The presence of Clustered Regularly Interspaced Short Palindromic Repeats (CRISPRs) was evaluated with CRISPRCasFinder (https://crisprcas.i2bc.paris-saclay.fr/CrisprCasFinder/Index (accessed on 2 May 2023)). DNA sequence analysis was performed with Artemis/ACT v17.0.1 (http://sanger-pathogens.github.io/Artemis/). The manual curated annotation of mobile genetic elements was carried out via a BLAST homology search of the following databases: the National Center for Biotechnology Information (NCBI) protein database (https://blast.ncbi.nlm.nih.gov/Blast.cgi?PAGE=Proteins (accessed on 9 May 2023)), Pfam (available under the InterPro consortium, https://www.ebi.ac.uk/interpro/search/sequence/ (accessed on 9 May 2023)), Virfam (http://biodev.cea.fr/virfam/ (accessed on 9 May 2023)) and Viridic (http://rhea.icbm.uni-oldenburg.de/VIRIDIC/ (accessed on 9 May 2023)). RUPs, BOX elements and SPRITEs were manually searched using BLAST (https://blast.ncbi.nlm.nih.gov/Blast.cgi?PAGE=Nucleotides (accessed on 2 May 2023)).

### 2.7. Bacterial Lysis for qPCR Analysis

Bacterial cells were grown at 37 °C in TSB broth until they reached the early exponential phase (OD_590_ = 0.2, roughly 5 × 10^8^ CFU/mL); then, 1 mL was harvested via centrifugation at 11,000 × *g* for 2 min. The cell pellets were dry vortexed and incubated at 37 °C in 30 µL of lysis solution (DOC 0.1%, SDS 0.008%) until clarification (about 10–15 min). The volume was then adjusted to 300 µL with TE 1×, pH 8.0 [18,19].

### 2.8. PCR, qPCR and Direct PCR Sequencing

PCR and direct PCR sequencing were carried out as previously described [27]. Quantitative real-time PCR was carried out with the KAPA SYBR FAST qPCR kit Master Mix Universal (2X) (Merck, Darmstadt, Germany) on a LightCycler 1.5 apparatus (Roche Diagnostics, Mannheim, Germany). The real-time PCR mixture contained, in a final volume of 20 μL, 1× KAPA SYBR FAST qPCR reaction mix, 5 pmol of each primer and 1 μL of bacterial lysate. The thermal profile was an initial 3 min denaturation step at 95 °C followed by 40 cycles of repeated denaturation (0 s at 95 °C), annealing (20 s at 60 °C) and polymerization (45 s at 72 °C). The temperature transition rate was 20 °C/s in the denaturation and annealing step and 5 °C/s in the polymerization step. Oligonucleotide primers and their properties are reported in Appendix A. Excised forms were detected and quantified using divergent primers targeting the ends of the elements, while the reconstitution of the integration sites was detected using primers targeting their flanking regions. A standard curve for the *gyrB* gene of *S. pneumoniae* BM6001 was built and used to standardize results. Melting curve analysis was performed to differentiate the amplified products from primer dimers.

### 2.9. Competent Cells Preparation and Transformation

*S. pneumoniae* competent cells’ preparation and transformation were carried out essentially as described in [28]. Briefly, precompetent pneumococcal cells were obtained by growing bacteria in TSB until the early exponential phase (OD_590_ = 0.1), then diluting them 100-fold in the Competence Transformation Medium (CTM) composed of TSB with 20% glucose, 4% BSA and 1% CaCl_2_. Cells were collected at 15 min intervals during the exponential phase of growth in CTM after 1 h of incubation at 37 °C. Mitomycin C was added 30 min after the dilution in CTM at a final concentration of 2 ng/mL. Competence curve was obtained by adding 1 µg/mL of transforming chromosomal DNA, carrying the *nov-1* point mutation, and 100 ng/mL of competence-stimulating peptide (CSP) to the precompetent cells. The transformation mixture was incubated at 37 °C for 45 min. The selection of transformants was obtained by a multilayer plating procedure in the presence of 10 µg/mL novobiocin [29]. The *S. pneumoniae* FP10 standard recipient was used as a control strain.

### 2.10. Phylogenetic Analysis and Nucleotide Homology Searching

The genome sequence data of *S. pneumoniae* isolates collected from 1916 to 1999 [30,31] were downloaded from the NCBI Nucleotide Database (https://www.ncbi.nlm.nih.gov/nuccore/ (accessed on 10 May 2023)) and the NCBI Sequence Read Archive (https://www.ncbi.nlm.nih.gov/sra/ (accessed on 10 May 2023)) and used for BM6001 phylogenetic analysis. A local database of the genomes was built using pp-sketchlib v2.1.1 with sketch v.2.1.1, k-mer lengths 13–29, a sketch size of 9984 and dense seeds (https://github.com/bacpop/pp-sketchlib). Population analysis was conducted with PopPUNK v2.6.0 (https://github.com/bacpop/PopPUNK) with the ‘--fit-model lineage’ parameter for data fitting using the (i) whole-genome sequences to estimate the overall genetic diversity and relatedness of isolates and (ii) core and accessory genome distances to identify specific lineages. PopPUNK outputs were visualized using the online epidemiology platform Microreact (https://microreact.org (accessed on 12 May 2023)). For nucleotide homology searching, the BM6001 mobilome was used as the BLAST query against assembled genomes. The SA41 and SA42 genomes were only available as raw sequencing data and were thus assembled into contigs using SPAdes genome assembler v.3.12.0 (https://github.com/ablab/spades) with default parameters.

### 2.11. Nucleotide Sequences Accession Numbers

The complete genome sequence of *Streptococcus pneumoniae* BM6001 is available under GenBank accession no. CP107038, whereas Nanopore and Illumina sequencing reads are available under Sequence Read Archive (SRA) accession no. SRR21857739 and SRR21857738, respectively.

## 3. Results

### 3.1. The Genome of S. pneumoniae BM6001

The sequence analysis showed that the BM6001 genome is organized in one circular chromosome of 2,293,748 base pairs (bp) in length, with an average GC content of 39.54% (Figure 1). The genome contains 2403 open reading frames (ORFs), distributed on the sense strand (1172) and on the antisense strand (1231). An annotation with the prediction of a biological function was possible for 2125 ORFs, including 12 ribosomal RNA (rRNA) genes grouped in four rRNA operons, 59 tRNA genes, of which 15 are not adjacent to rRNA operons, and 3 structural RNAs, namely the (i) tRNA-like/mRNA-like RNA, (ii) the signal recognition particle RNA and (iii) the ribonuclease P RNA. BM6001 genome harbors a type 19F capsule locus located between *dexB* (cds spanning nts 386,479 to 388,086) and *aliA* (cds spanning nts 405,911 to 407,893) as in almost all pneumococcal strains [2,32,33]. A total of 15 capsule genes were predicted to be a single transcriptional unit with a promoter-like sequence located upstream of *wzg/cps19A*, as reported in [2,32]. Between the last capsule gene *rmlD/cps19O* and *aliA*, the locus harbors the UDP-galactopyranose mutase *glf* gene, as reported for other serotypes, including 19C [2]. The *pspC* locus, encoding the multifunctional pneumococcal surface protein C (PspC) in BM6001, contains two tandem copies of the *pspC* gene spaced by an IS*1167* element, as already described in other *S. pneumoniae* clinical isolates [3]. The 1764 bp *pspC* copy (cds spanning nts 2,239,644 to 2,241,407) encodes the novel PspC9.5 variant containing an LPXTG anchoring domain, whereas the 2172 bp *pspC* copy (cds spanning nts 2,243,070 to 2,245,241) encodes the novel PspC3.14 variant containing the classical pneumococcal choline-binding domain. The *comCDE* competence locus spans nts 2,288,375 to 2,290,595 and contains the *comC1* and *comD1* alleles [34,35,36], while the phase-variable type I restriction modification system *spnIII*, predominantly arranged as variant C, spans nts 529,953 to 538,028 [37].

### 3.2. The Mobilome of BM6001

The BM6001 mobilome accounts for 15.54% (356,521 bp) of the whole genome and includes (i) the composite ICE Tn*5253* spanning nts 1,126,687 to 1,191,345, already described elsewhere [14,15,16,17,18,19]; (ii) the novel IME Tn*7089*; (iii) the novel transposon Tn*7090*; (iv) 3 prophages and 2 satellite prophages; (v) 5 genomic islands (GIs); and (vi) 72 insertion sequences (ISs), making up 2.8% (64,198 bp) of the genome and belonging to 11 different families (Appendix A). BM6001 also harbors 69 RUPs (49 being subtype A, 10 being subtype B1, 7 being subtype B2 and 3 being subtype C) [38]; 153 BOX elements, of which 14 are composed of subunit A and B and 59 of subunits B and C [39]; and 31 SPRITEs, of which 7 are flanked by terminal inverted repeats [40]. No plasmids were detected.

### 3.3. IME Tn7089

The novel IME Tn*7089* is 9089 bp in length, spans nts 339,362 to 348,450 and is characterized by an overall GC content of 30.48%. The element contains 11 ORFs, which are all transcribed in the same direction (Figure 2). Manual homology-based annotation attributed a putative function to only 6 out of 11 ORFs (Table 1). Tn*7089* contains (i) an integration/excision module, constituted by the *int* and *xis* genes, coding for an integrase and an excisionase; (ii) *orf3* and *orf7* coding for a Rep protein and an FtsK homologous protein likely involved in the mobilization; and (iii) *orf4* and *orf5* coding for metal-dependent phosphoesterase proteins. The latter of these may be involved in the hydrolysis of the pyrophosphate released during nucleotide polymerization, and thus may influence the kinetics of DNA synthesis [41]. The NCBI database of 90,819 complete microbial genomes (accessed in May 2023) was interrogated using the Tn*7089* DNA sequence as a query. Tn*7089*-like elements were found in 11 *S. pneumoniae* genomes and contained nt changes up to 2.56% of the sequence length (Figure 2). In two additional genomes, a Tn*7089*-like element contains a 963 bp deletion and a 2682 bp insertion at the 3′ end. Furthermore, a Tn*7089*-like element containing a 1639 bp DNA deletion and an 870 bp DNA insertion at the 3′ end was found in the *Streptococcus mitis* B6 genome.

### 3.4. Transposon Tn7090

The novel 9053 bp long (nts 1,920,769 to 1,929,821) transposon Tn*7090* has a GC content of 30.78% and is flanked by two 460 bp long direct repeats homologous to the IS*Spn7* element of the IS*5* family (Figure 3). Tn*7090* contains seven ORFs organized as a single transcriptional unit (Table 2). Orf1 is predicted to be a divalent cation-dependent glycerophosphodiester phosphodiesterase family protein which hydrolyzes glycerophosphodiesters into sn-glycerol 3-phosphate and alcohol [42]. In bacteria, this protein family is involved in virulence and adhesion to host cells, as demonstrated for *Haemophilus influenzae* protein D [43]. Orf2 and Orf7 are predicted to be extracellular solute-binding proteins, whereas Orf3-Orf4 to constitute an ATP-binding cassette transporter. Orf5 is homologous to proteins of the MgtC/SapB/SrpB/YhiD family, likely involved in intracellular survival and Mg^2+^ binding [44]. Orf6 is homologous to a phosphatase of the haloacid dehalogenase-like hydrolase family [45]. Altogether, these genes are predicted to code for proteins likely involved in the uptake and binding of Mg^2+^ cations in the adhesion to host cells and intracellular survival. The homology search analysis showed that Tn*7090*-like elements are present in 43 out of 153 complete genomes of *S. pneumoniae*.

### 3.5. Prophages

The BM6001 genome search carried out with PHASTER software led to the identification of three prophages, ΦBM6001.1, ΦBM6001.2 and ΦBM6001.3, ranging in size from 36,108 bp to 42,445 bp, and two satellite prophages, ΦBM6001.4 and ΦBM6001.5 (Figure 4). Prophages have a GC content similar to the host, while satellite prophages have a lower GC content [11]. Virfam analysis and a BLAST homology search, conducted in public protein databases and Pfam, predicted the presence of genes encoding phage structural proteins and lytic enzymes such as holin and amidase for the three prophages. Genes encoding proteins involved in the DNA replication process and in lysogeny were found in both prophages and satellite prophages (Appendix A). Only prophage ΦBM6001.2 contains virulence determinants, namely *orf31*, predicted to code for a YopX homolog [46], and *orf38*, predicted to encode the “virulence-associated protein E”, which is widely spread among pneumococcal phages and associated with *S. pneumoniae* virulence in a mouse pneumonia model [11] and a tyrosine-tRNA gene, which might contribute to modulating the tRNA pools of bacteria, improving the translation efficiency of viral genes [47]. The homology search showed that ΦBM6001.1 is essentially identical to the *S. pneumoniae* siphovirus prophage IPP42 in the AP200 genome [10]. ΦBM6001.2 is homologous to *S. pneumoniae* siphovirus prophage phiARI0131-1 [48] except for the presence of a rearranged copy of the structural gene *orf6* and the acquisition of a 1202 bp DNA segment (*orf29* to *orf33*). ΦBM6001.3 has a mosaic structure where the 3′ end lysogeny module (*orf40* to *orf49*) is similar to that of *S. pneumoniae* siphovirus prophage IPP69, whereas the remaining sequence is homologous to *S. pneumoniae* siphovirus prophage MM1 [10]. Finally, satellite prophages ΦBM6001.4 and ΦBM6001.5 are essentially identical to satellite prophages Javan 759 and Javan 757, respectively [11].

### 3.6. Genomic Islands

Strain-specific genomic DNA fragments have been described in the genome of *S. pneumoniae* and were named (i) gene clusters [49,50], (ii) regions of diversity (RD) [51,52], or (iii) accessory regions [8,22]. These terms were used as synonyms. In addition, two DNA fragments were described as genomic islands, namely pneumococcal pathogenicity island 1 (PPI1) [20,21] and the PTS island [53]. Using the IslandViewer and BLAST software b, we identified five specific DNA fragments, ranging in size from 5188 bp to 22,489 bp, flanked by direct repeats (7 to 15 bp in length) and characterized by a lower GC content than the average of BM6001 chromosome. Since these DNA fragments are neither prophages nor ICEs/IMEs, we refer to them as genomic islands (GIs) (Figure 5). GI-BM6001.1 is located in the intergenic region between the cell division *divIB* gene and the orotidine-5′-phosphate decarboxylase *pyrF* gene. GI-BM6001.1 is a variant of TIGR4 RD5 [51] and contains four ORFs, including *orf2*, which is predicted to code for an HesA/MoeB/ThiF family enzyme that catalyzes the adenylation of ThiS, a sulfur carrier protein involved in the biosynthesis of thiamine [54], and *orf3* and *orf4*, coding for an ATP-binding cassette transporter. Furthermore, a truncated IS of the IS*30* family is present at the 3′ end. GI-BM6001.2, flanked by a 14 bp inverted repeat, is a variant of the pathogenicity island PPI1 [21] located at the same intergenic region as that found in the TIGR4 genome. Compared to TIGR4 PPI1, a 5657 bp insertion produces a 181 bp deletion involving the PPI1 *sp1038* gene, while at the 3′ end, a 1715 bp insertion produces a 13,297 bp deletion, corresponding to PPI1 *sp1048*-*sp1066*. GI-BM6001.3 is a variant of the R6 Cluster 4 [50] and is part of a variable locus in *S. pneumoniae* genomes, located between the conserved NADP-specific glutamate dehydrogenase *gdhA* and 50S ribosomal protein L7/L12 *rplL* genes (Figure 5) [55,56]. The island is flanked by a 7 bp DR and contains a truncated copy of IS*Smi2* of the IS*1182* family at the 5′ end. The DR likely acts as a hotspot for integration, as a truncated copy of the IS*5* element was found downstream of the 3′ end of GI-BM6001.3, and a complete copy of IS*5*, flanked by the 7 bp DR, was found in other pneumococcal genomes. GI-BM6001.3 contains 11 ORFs, of which, *orf4* to *orf11* are predicted to be a single transcriptional unit likely involved in the metabolism of sialic acid [57]. Orf4 is a phosphatidylglycerophosphatase A, Orf9-Orf8-Orf7-Orf6 constitute an ATP-binding cassette transporter, Orf10 is a cyclically permuted mutarotase family protein and Orf11 is homologous to an N-acetylmannosamine-6-phosphate 2-epimerase (Appendix A). GI-BM6001.4 is flanked by 15 bp imperfect direct repeats containing the 3′ end of the *guaA* gene. The island carries nine ORFs including *orf3*, a cell division FtsK gene and *orf1* and *orf2*, predicted to encode an integration/excision module. GI-BM6001.4-like sequences were only found in two pneumococcal complete genomes and in other streptococcal genomes such as *Streptococcus suis, Streptococcus equi* and *Streptococcus agalactiae*. GI-BM6001.5 is a variant of the PTS island described in *S. pneumoniae* Hungary 19A-6 and OXC141 genomes [53] located in an intergenic region. Compared to the PTS island, GI-BM6001.5 contains the insertion of IS*Spn5* in an intergenic region and an insertion of *orf7* within the putative cellobiose PTS system module.

### 3.7. Excision and Attachment Sites’ Reconstitution of BM6001 MGEs

PCR and sequencing analysis were used to detect and quantify the restored insertion sites and the excised forms of the different genetic elements composing the BM6001 mobilome. As already reported, ICE Tn*5253* excises from bacterial chromosomes, producing circular forms and the reconstitution of the *att*B target site [18]. IME Tn*7089* is able to excise from the bacterial chromosome, producing a circular form where the left and right ends are joined by a 52 bp sequence (*att*Tn). The excision of the element reconstitutes the chromosomal 52 bp Tn*7089* attachment site (*att*B), whereas upon chromosomal integration, the element is flanked by *att*L and *att*R, identical to *att*Tn and *att*B, respectively. *att*L-*att*Tn contain four nt changes compared to *att*R-*att*B. *att*B corresponds to the last 52 nts of the *rpsI* coding sequence encoding the 30S ribosomal protein S9. In liquid culture of BM6001, circular forms of Tn*7089* were present at a concentration of 3.92 × 10^−8^ (±1.08 × 10^−8^) copies per chromosome, whereas the reconstituted *att*B site was at 5.73 × 10^−8^ (±5.78 × 10^−9^) copies per chromosome (Table 3). Also, Tn*7090* is able to excise from the chromosome and produce a circular form. Upon excision, a copy of IS*Spn7* remains in the chromosome, while a copy of IS*Spn7* joins the left and right ends of the transposon circular forms. Tn*7090* circular forms were present at a concentration of 4.19 × 10^−4^ (±1.60 × 10^−4^) copies per chromosome, whereas the reconstituted *att*B site was at 3.21 × 10^−4^ (±3.11 × 10^−5^) copies per chromosome (Table 3). Prophages and satellite prophages also produce excised forms where the left and right ends are joined by an *att*P sequence, restoring the relative *att*B integration sites. Prophages and satellite prophage *att*B sites are insertional hotspots, since other genetic elements integrate at the same site [11,58]. In ΦBM6001.1, the 21 bp *att*B is identical to *att*P and is located in the intergenic region, spacing the adenylosuccinate synthase gene and tRNA-specific adenosine deaminase *tadA* gene (Figure 4). ΦBM6001.2 is flanked by 14 bp long *att*L and *att*R, which differ by one nt change. *att*B is located in the intergenic region between the NAD(P)/FAD-dependent oxidoreductase gene and the DNA-binding protein *whiA* gene. ΦBM6001.3 is flanked by 11 bp long *att*L and *att*R, identical to *att*B and *att*P, respectively, and differing by two nt changes. The ΦBM6001.3 *att*B corresponds to 11 nts (2,042,299 to 2,042,309) located at the 5′ end of the *comGC/cglC* late competence gene [59,60,61]. Both ΦBM6001.4 and ΦBM6001.5 satellite prophages integrate in intergenic regions, upstream of *ychF* and *uvrA* genes, respectively. ΦBM6001.4 *att*P and *att*B sites are 26 bp long with *att*L-*att*B differing from *att*R-*att*P by four nt changes, whereas ΦBM6001.5 *att*P and *att*B sites are 24 bp long with *att*L-*att*P differing from *att*R-*att*B by one nt change. Excised forms of prophages and satellite prophages range from 1.12 × 10^−5^ to 2.35 × 10^−1^, while the frequency of reconstituted integration sites varies from 3.23 × 10^−7^ to 3.01 × 10^−2^ (Table 3). Finally, neither circular forms nor reconstituted chromosomal target sites could be detected for genomic islands.

### 3.8. Impact of ΦBM6001.3 on Genetic Transformation

ΦBM6001.3 chromosomal insertion disrupts the major pilin-like *comGC*/*cglC* gene, a member of the *comG*/*cgl* operon involved in the formation of the DNA uptake machinery during transformation [60,62]. To investigate if ΦBM6001.3 impairs transformation [48], BM6001 competent cells were collected over a 90 min interval and transformed with a donor DNA carrying a point mutation for a selectable marker [28]. Competent cells were also obtained in the presence of a mitomycin C stimulus. In our experimental conditions, we were not able to obtain transformants at any time point sampled, neither with nor without mitomycin C (<1.07 × 10^−7^ transformants/total CFU). Then, we quantified the ΦBM6001.3 excised forms and *att*B restoration frequency in bacterial lysates obtained from competent cells. Over time, we did not observe a significant variation in the frequencies of ΦBM6001.3 excised forms and reconstituted *att*B in the untreated competent cells (average values of 8.93 × 10^−1^ and 4.32 × 10^−2^, respectively; Figure 6), whereas mitomycin C induced a 10-fold frequency increase compared to the untreated control after 1 h of treatment.

### 3.9. Phylogenetic Analysis of BM6001

To investigate whether the lack of competence for genetic transformation drives the accumulation of multiple MGEs within the BM6001 genome, a collection of sequenced historical pneumococcal genomes was used to analyze the phylogenetic relationship of BM6001 [30,31]. A total of 195 genomes were used to generate a phylogenetic tree using both the core and accessory genome information (Figure 7). BM6001 collocates in the same lineage with five other genomes, namely GA13455, CDC1873-00, GA04375, SA41 and SA42. The core genome distances vary between 0.005647123 and 0.024928689, while the accessory genome distances are between 0.14985132 and 0.4647357 (Appendix A), indicating that most changes are clustered in the mobile portion of the genome. The 292,323 bp representing the BM6001 mobilome devoid of ISs were used as a probe for the homology search in the five genomes belonging to the same lineage. Regions of homology were identified in all the genomes, ranging from a minimum of 34,436 bp (11.8%) in GA04375 to a maximum of 129,463 bp (44.3%) in SA42.

## 4. Discussion

In this study, complete genome sequence analysis allowed the definition of the BM6001 mobilome, which includes, beyond the well-characterized ICE Tn*5253*, (i) 2 novel elements, IME Tn*7089* and transposon Tn*7090*; (ii) 3 prophages and 2 satellite prophages; (iii) 5 GIs; (iv) several ISs, RUPs, 153 BOX elements and SPRITEs. MGEs represent 15.54% of the BM6001 genome and prophages and satellite prophages constitute 6.49% of the genome alone, while the average for pneumococci is 3.8% [11]. The annotation of the GI genes revealed that they contained five predicted transport systems, which are possibly involved in pneumococcal virulence [8,20,21]. Furthermore, ΦBM6001.2 contains virulence determinants coding for the YopX family protein and the virulence-associated E family protein [11], while Tn*7090* codes for proteins likely involved in the uptake and binding of Mg^2+^ cations, in the adhesion to host cells and in intracellular survival. It has been shown that ICE Tn*5253* is capable of excision from the BM6001 chromosome, circularization and subsequent conjugal transfer to other pneumococcal strains [18]. Instead, BM6001 does not acquire exogenous DNA via genetic transformation even though *S. pneumoniae* are naturally transformable species. Given this, we first hypothesized that the insertion of ΦBM6001.3 within the major pilin-like *comGC*/*cglC* gene impairs the DNA entry, since the gene is part of the *comG/cgl* operon involved in the formation of the pneumococcal DNA uptake machinery during transformation. It is noteworthy that ΦBM6001.3 excises at a high frequency from the pneumococcal chromosome, restoring the insertion site in about 3% of the cells, which therefore have an intact copy of *comGC*/*cglC*. Mitomycin C induction further increases ΦBM6001.3’s frequency of excision and restoration of the intact *comGC*/*cglC* gene of about one order of magnitude. Nevertheless, BM6001 could not be transformed in any of our experimental conditions. As observed for ΦBM6001.3, mitomycin C treatment induces the excision of the lysogenic phage Φ1207.3 [63,64,65,66], which inserts within the pneumococcal *comEC*/*celB* competence gene impairing genetic transformation and increases the free locus restoration frequency. Differently from ΦBM6001.3, this restores a low level of competence for genetic transformation (9.96 × 10^−4^ transformants/total CFU). To investigate whether other factors could explain the lack of transformability of BM6001, we compared the BM6001 pneumococcal competence genes to those of TIGR4 and found that all of them were essentially conserved, except for comGC/cglC disruption. Furthermore, the presence of restriction modification (RM), toxin–antitoxin (TA) and abortive infection (Abi) systems, carried by BM6001 MGEs, could concur with transformation impairment; in fact, Tn*5253* harbors two copies of the *pezAT* TA system, which is known to impair transformation [67], a methylase and an Abi system, while ΦBM6001.2 carries a restriction enzyme and an Abi protein. Transformation impairment is probably the result of multiple strategies adopted by the BM6001 mobilome to avoid its curing; in fact, genome-wide analysis has shown that transformation can lead to the removal of inserted MGEs via homologous recombination and that many MGEs tend to disrupt competence genes to block transformation and consequently avoid this phenomenon [48,68]. The genomic comparison showed that BM6001 is in the same lineage of five other historical strains, but it only shares less than 50% of its mobilome sequences with the closest relative. Altogether, our findings suggest that BM6001 has progressively accumulated many MGEs while losing competence for genetic transformation, to become a sort of genomic *cul-de-sac*.

## Figures and Tables

**Figure 1 microorganisms-11-01646-f001:**
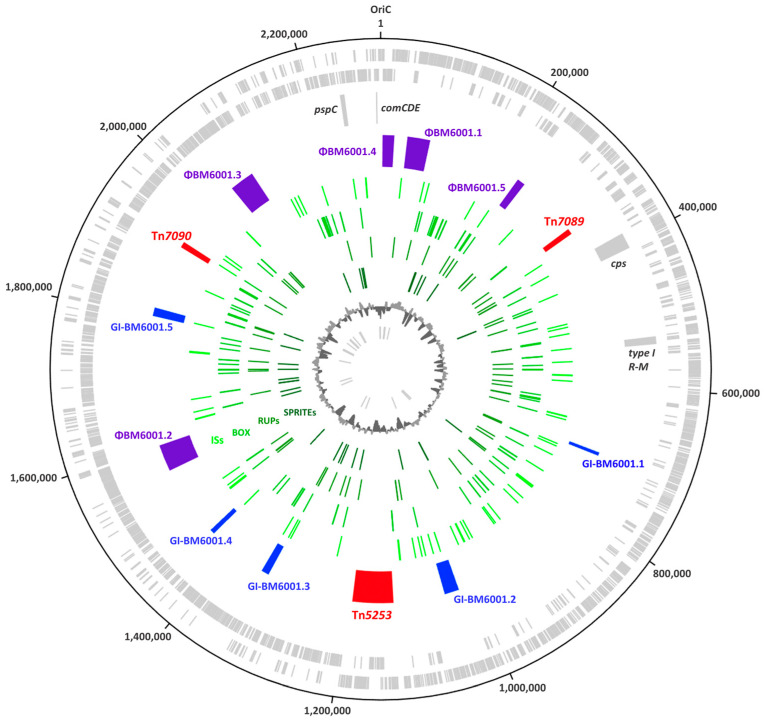
Circular representation of the *S. pneumoniae* BM6001 genome. Circle 1 (outer) and circle 2 show the predicted coding regions located on the plus and minus strands, respectively. In the third circle, the following loci are reported as gray blocks: capsule (*cps*), type I restriction modification system (type I R-M), pneumococcal surface protein C (*pspC*) and *comCDE* competence locus. Circle 4 reports IME Tn*7089*, ICE Tn*5253* and transposon Tn*7090* as red blocks; prophages ΦBM6001.1, ΦBM6001.2 and ΦBM6001.3 and satellite prophages ΦBM6001.4 and ΦBM6001.5 as purple blocks; genomic islands GI-BM6001.1, GI-BM6001.2, GI-BM6001.3, GI-BM6001.4 and GI-BM6001.5 as blue blocks; the 72 insertion sequences, the 153 BOX elements, the 69 RUPs and the 31 SPRITEs are represented as green ticks in circles 5, 6, 7 and 8, respectively. The ninth circle shows GC content. The innermost circle indicates RNA genes. The image was created using Artemis DNA-Plotter (v.17.0.1).

**Figure 2 microorganisms-11-01646-f002:**
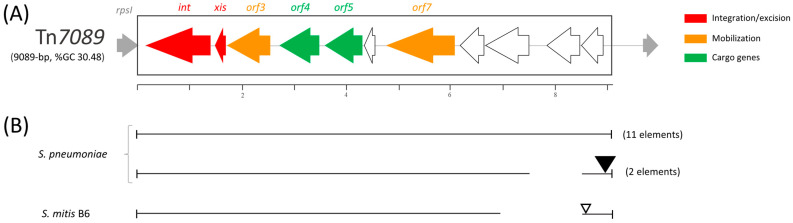
(**A**) Structure of *S. pneumoniae* IME Tn*7089* and (**B**) schematic comparison of Tn*7089* with other Tn*7089*-like elements. (**A**) Tn*7089* is a 9089 bp long IME and contains 11 ORFs, including genes for integration/excision, mobilization and cargo genes. ORFs and their direction of transcription are represented by arrows; annotated ORFs are indicated. The scale is in kilobases. (**B**) Tn*7089*-like elements were found in 13 *S. pneumoniae* complete genomes and in the genome of *S. mitis* strain B6. Regions of homology are indicated by lines, whereas DNA insertions are depicted by triangles. Sequence length varies from 8320 bp of *S. mitis* B6 element to 11,203 bp of the elements found in 2 different *S. pneumoniae* strains.

**Figure 3 microorganisms-11-01646-f003:**
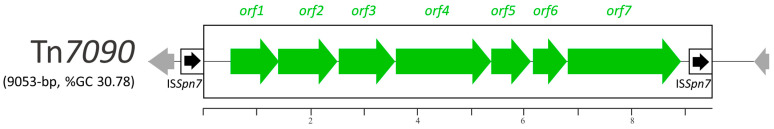
Structure of *S. pneumoniae* transposon Tn*7090*. Tn*7090* contains a single transcriptional unit likely involved in the uptake and binding of Mg^2+^ cations, in the adhesion to host cells and in intracellular survival. The element is flanked by truncated copies of IS*Spn7*. ORFs and their directions of transcription are represented by arrows. Insertion sequences are reported as thinner boxed arrows. Chromosomal genes are represented by thin gray arrows. The scale is in kilobases.

**Figure 4 microorganisms-11-01646-f004:**
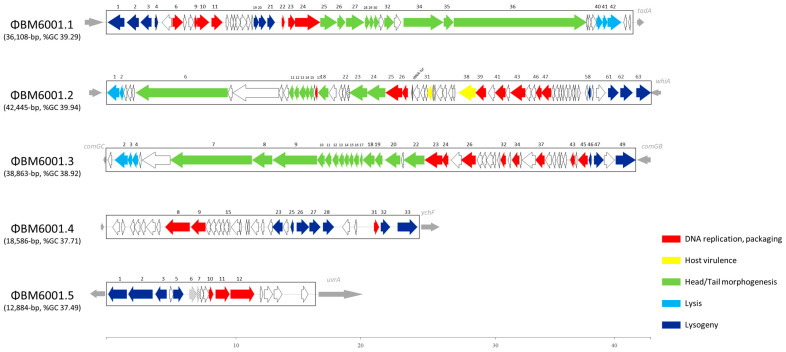
Schematic structures of *S. pneumoniae* BM6001 prophages. ΦBM6001.1, spanning nts 42,045 to 78,152, is essentially identical to the *S. pneumoniae* siphovirus prophage IPP42. ΦBM6001.2, nts 1,556,579 to 1,599,023, is homologous to the pneumococcal siphovirus prophage phiARI0131-1. ΦBM6001.3, nts 2,042,310 to 2,081,172, has a mosaic structure where the 3′ end lysogeny module (*orf40* to *orf49*) is similar to that of *S. pneumoniae* siphovirus prophage IPP69, whereas the remaining sequence is similar to *S. pneumoniae* siphovirus phage MM1. ΦBM6001.4, nts 2,961 to 21,546, is homologous to prophage Javan 759, and ΦBM6001.5, nts 228,793 to 241,676, is essentially identical to Javan 757. Via manual homology-based annotation, a putative function was only attributed to a minority of prophage ORFs ranging from 38% to 57% (Table 2). ORFs and their direction of transcription are represented by arrows, and annotated ORFs are indicated by their number. Different colors highlight the different functional gene categories. Chromosomal genes flanking the prophage insertion sites are represented by thin gray arrows. The scale is in kilobases.

**Figure 5 microorganisms-11-01646-f005:**
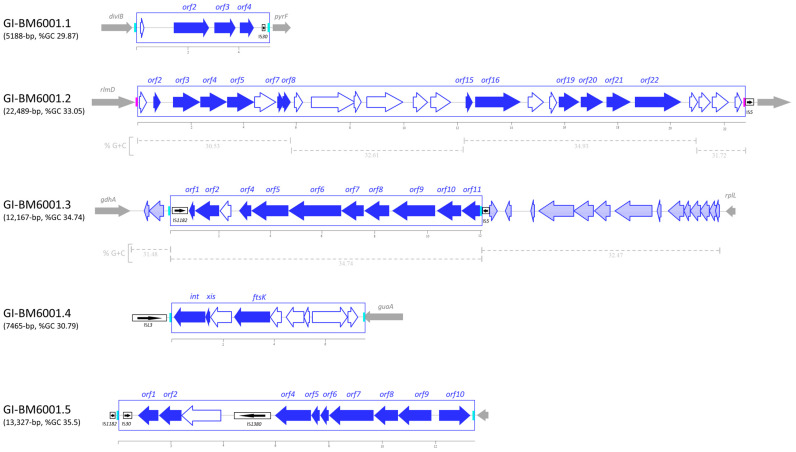
Structure of *S. pneumoniae* BM6001 genomic islands (GIs). GI-BM6001.1 is a variant of the TIGR4 RD5 region of diversity, spans nts 706,529 to 711,716 and is flanked by 16 bp imperfect direct repeats (DRs). GI-BM6001.2, spanning nts 1,022,483 to 1,044,971 and flanked by a 14 bp inverted repeat (IR), is a variant of the TIGR4 pathogenicity island 1. Upon a 5657 bp insertion (*orf9* to *orf13*) and a 1715 bp insertion (*orf23* to *orf25*), we could identify four regions differing in GC content. GI-BM6001.3 is a variant of the R6 Cluster 4, spans nts 1,329,412 to 1,341,578, is part of an *S. pneumoniae* genome variable locus located between *gdhA* and *rplL* and is characterized by a higher GC content compared to the remaining fragments of the locus. GI-BM6001.4 spans nts 1,436,320 to 1,443,784 and is integrated at the 3′ end of *guaA*, leading to a 15 bp duplication of the target site, restoring the coding sequence. GI-BM6001.5 is a variant of the PTS island, spanning nts 1,804,197 to 1,817,523, characterized by the insertions of IS*Spn5* and of *orf7*, which integrates in the cellobiose PTS system module (*orf4* to *orf9*). ORFs and their direction of transcription are represented by arrows; annotated ORFs are indicated. Insertion sequences are reported as black boxed arrows. DRs and IRs are indicated by light blue and pink boxes, respectively. Chromosomal genes flanking the genomic island insertion sites are represented by thin gray arrows.

**Figure 6 microorganisms-11-01646-f006:**
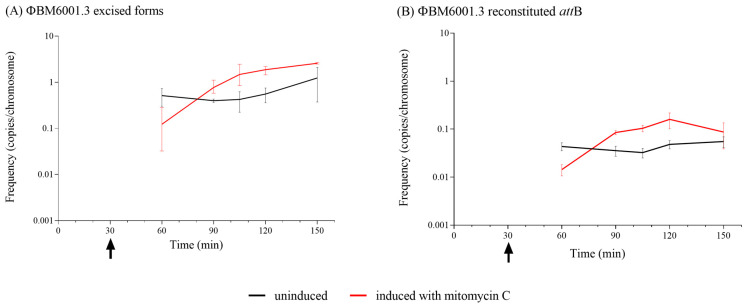
ΦBM6001.3 excised forms and reconstituted *att*B frequencies in BM6001 competent cells. ΦBM6001.3 excised forms (**A**) and reconstituted *att*B (**B**) frequencies were determined at different time points using BM6001 cells either treated with mitomycin C (2 ng/mL) or untreated. Treatment with mitomycin C was performed 30 min after dilution in CTM at time 0, as indicated by arrows. No significant variation was observed in the frequency of ΦBM6001.3 excised forms and reconstituted *att*B in the untreated BM6001 competent cells, whereas in the presence of mitomycin C stimulus, a 10-fold frequency increase was observed over time. Results are reported as means and standard errors resulting from 2 to 6 technical replicates from 2 independent experiments.

**Figure 7 microorganisms-11-01646-f007:**
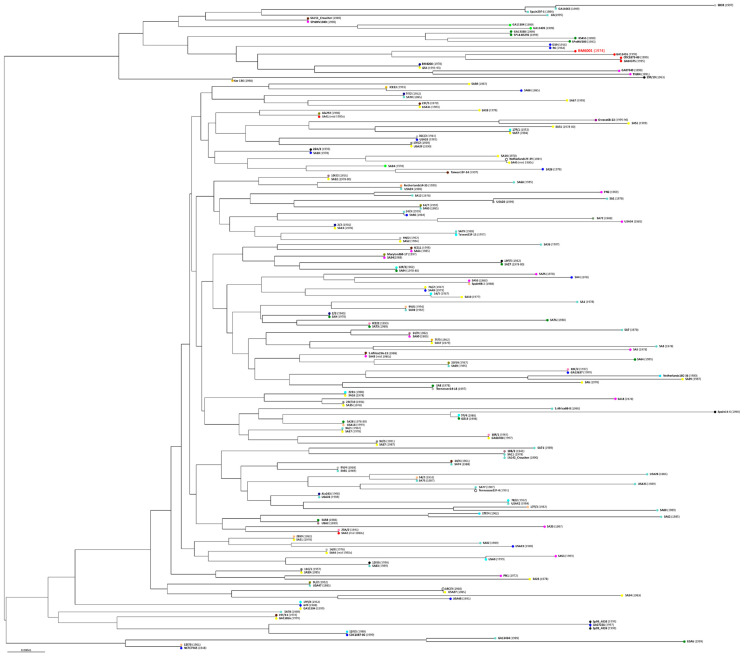
Phylogenetic tree of BM6001 and 195 *S. pneumoniae* isolates collected from 1916 to 1999. Genetic relatedness was evaluated with the PopPUNK tool using the ‘--fit-model lineage’ parameter for data fitting. The phylogenetic tree was generated on the sequence data available at GenBank with branch lengths, calculated based on k-mer analysis, indicating the number of nucleotide substitutions per site (scale bar), whereas 23 lineages (colored dots) were obtained including both core and accessory genome sequence information. Lineage 9 includes BM6001 and strains GA13455, CDC1873-00, GA04375, SA41 and SA42 (red dots). Names of the strains are indicated, and the date of isolation is reported in parenthesis.

**Table 1 microorganisms-11-01646-t001:** Annotated ORFs of IME Tn*7089*.

ORF (aa) *^a^*	Predicted Protein	Homologous Protein ID/Origin Identity (%) [E Value] *^b^*	Pfam Domain *^c^* (aa) [E Value]
*orf1* (406)	Tyrosine-type DNA integrase		Integrase, catalytic domain (210–387) [3.2 × 10^−15^]
*orf2* (89)	Excisionase, putative		Helix–turn–helix domain, group 17 (42–86) [7.6 × 10^−11^]
*orf3* (264)	Rep protein, putative	CBJ21467.1/*S. mitis* 152/261 (58%) [3.0 × 10^−111^]	
*orf4* (257)	Metal-dependent phosphoesterase	CBJ21468.1/*S. mitis* 234/254 (92%) [1.0 × 10^−179^]	Calcineurin-like phosphoesterase domain, ApaH type (1–194) [3.9 × 10^−9^]
*orf5* (242)	Metal-dependent phosphoesterase		Calcineurin-like phosphoesterase domain, ApaH type (4–196) [4.8 × 10^−12^]
*orf7* (439)	FtsK/SpoIIIE domain-containing protein		FtsK domain (188–371) [1.1 × 10^−14^]

*^a^* The number of amino acids of the predicted protein is shown in parenthesis. *^b^* Determined via compositional matrix adjustment. *^c^* Numbers in parentheses represent the part of the protein homologous to the Pfam domain.

**Table 2 microorganisms-11-01646-t002:** Annotated ORFs of Tn*7090* transposon.

ORF (aa) *^a^*	Predicted Protein	Homologous Protein ID/Origin Identity (%) [E value] *^b^*	Pfam Domain *^c^* (aa) [E value]
*orf1* (286)	Glycerophosphodiester phosphodiesterase		Glycerophosphodiester phosphodiesterase domain (18–102) [4.0 × 10^−7^]
*orf2* (353)	Bacterial extracellular solute-binding protein	WP_000475761.1/*Streptococcus pneumoniae* 353/353 (100%) [0.0]	Bacterial extracellular solute-binding domain (88–338) [4.3 × 10^−28^]
*orf3* (334)	ABC transporter ATP-binding protein		ABC transporter-like, ATP-binding domain (18–160) [2.6 × 10^−36^]
*orf4* (564)	Iron ABC transporter permease		ABC transporter type 1, transmembrane domain MetI-like (60–266) [2.9 × 10^−7^]; ABC transporter type 1, transmembrane domain MetI-like (375–551) [1.4 × 10^−12^]
*orf5* (233)	MgtC/SapB/SrpB/YhiD family protein		MgtC/SapB/SrpB/YhiD domain (17–140) [5.0 × 10^−35^]
*orf6* (199)	Haloacid dehalogenase-like hydrolase		Haloacid dehalogenase-like hydrolase (3–182) [1.8 × 10^−13^]
*orf7* (675)	Bacterial extracellular solute-binding protein		Bacterial extracellular solute-binding protein (334–619) [7.1 × 10^−17^]; LacI-type HTH domain (1–66) [1.4 × 10^−11^]
*tnp* (133)	IS*Spn7*, family IS*5*, truncated	WP_001105201.1/*Streptococcus pneumoniae* 36/267 (29%) [5.0 × 10^−23^]	

*^a^* The number of amino acids of the predicted protein is shown in parenthesis. *^b^* Determined via compositional matrix adjustment. *^c^* Numbers in parentheses represent the part of the protein homologous to the Pfam domain.

**Table 3 microorganisms-11-01646-t003:** Real-time PCR quantification of BM6001 mobile genetic elements’ circular forms and reconstituted integration sites.

Mobile Genetic Element	Circular Forms *^a^*	Reconstituted Integration Site *^a^*
Tn*7089*	3.92 × 10^−8^ (±1.08 × 10^−8^)	5.73 × 10^−8^ (±5.78 × 10^−9^)
Tn*7090*	4.19 × 10^−4^ (±1.60 × 10^−4^)	3.21 × 10^−4^ (±3.11 × 10^−5^)
ΦBM6001.1	1.27 × 10^−3^ (±9.86 × 10^−4^)	4.12 × 10^−3^ (±4.84 × 10^−4^)
ΦBM6001.2	8.34 × 10^−3^ (±9.78 × 10^−3^)	4.74 × 10^−3^ (±3.5 × 10^−4^)
ΦBM6001.3	2.35 × 10^−1^ (±1.92 × 10^−1^)	3.01 × 10^−2^ (±2.17 × 10^−2^)
ΦBM6001.4	2.34 × 10^−2^ (±2.28 × 10^−3^)	1.12 × 10^−2^ (±1.49 × 10^−2^)
ΦBM6001.5	1.12 × 10^−5^ (±2.43 × 10^−6^)	3.23 × 10^−7^ (±1.35 × 10^−8^)

*^a^* Concentration was expressed as copies per chromosome. Results are reported as means and standard deviation resulting from 2 to 4 technical replicates.

## Data Availability

All data included in this study are deposited in public databases.

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
