# Peer review of "The Mobilome-Enriched Genome of the Competence-Deficient Streptococcus pneumoniae BM6001, the Original Host of Integrative Conjugative Element Tn5253, Is Phylogenetically Distinct from Historical Pneumococcal Genomes"

_microorganisms, 2023, doi:10.3390/microorganisms11071646_

Round 1

Reviewer 1 Report

The authors report a mobilome analysis of a unique pneumococcal strain following long read sequencing, and verify the ability for several of the mobile genetic elements to excise from the genome with qPCR analysis.  Overall the study is sound, though there are a few points in the text where the authors overstate their claims, as detailed below.

-Line 266: the authors do not show that these ORFs are transcribed in the same direction. Change the language to "predicted to be all transcribed"

-Line 297: same as above. Change the language to "predicted to be a single transcriptional unit" or show a Northern blot confirming.

-Line 298: the function of Orf1 is not confirmed in this study. Change language to "predicted to be a..."

-Lines 301-304: same as above. Change language to indicate that these are predicted protein functions as they are not confirmed in this study.

-Lines 388-391: same as above. Change language to indicated predicted single transcriptional unit and predicted functions of proteins

The details of bacterial growth conditions need to be explained further. Lines 83-85 discuss only liquid culture. Did the authors never grow the bacteria on agar plates? Further, what atmosphere was used for liquid growth? at higher ODs reported I would expect a microaerophilic atmosphere to be needed, or the addition of catalase/oxyrase to the liquid media to prevent bacterial death.

Lines 44-46 do not support the current understanding of the pneumococcal pangenome.  Only 1/3 of the genome is typically shared between strains, not 70-80%, and the majority of variation is in the accessory genes expressed, not allelic variation in core genes.

Lines 219-221 do not make sense, if about half the genes are on the sense and the antisense strands, what is meant by "the majority are transcribed in the same direction as DNA replication"?

Lines 332-334: Provide a description for the virulence functions of YopX and E family proteins in pneumococci

Lines 379-380 do not make sense, how does an insertion produce a deletion?

Lines 487-488: this hypothesis should also be introduced earlier in the paper, this provides an interesting scientific rationale for the study, which this reviewer did not appreciate until the end of the paper. More cursory readers might miss this interesting thought question instead thinking of this as a mere genome report.

Line 536-537: please include this unpublished data.

Figure 4 has a misspelling of lysogeny

Author Response

Thank you for the accurate review and understanding of our manuscript and for the helpful suggestions that allowed us to greatly improve the manuscript. 

Response to Reviewer 1 Comments

The authors report a mobilome analysis of a unique pneumococcal strain following long read sequencing, and verify the ability for several of the mobile genetic elements to excise from the genome with qPCR analysis.  Overall the study is sound, though there are a few points in the text where the authors overstate their claims, as detailed below.

-Line 266: the authors do not show that these ORFs are transcribed in the same direction. Change the language to "predicted to be all transcribed"

THE SENTENCE WAS MODIFIED AS SUGGESTED

-Line 297: same as above. Change the language to "predicted to be a single transcriptional unit" or show a Northern blot confirming.

THE SENTENCE WAS MODIFIED AS SUGGESTED

-Line 298: the function of Orf1 is not confirmed in this study. Change language to "predicted to be a..."

THE SENTENCE WAS MODIFIED

-Lines 301-304: same as above. Change language to indicate that these are predicted protein functions as they are not confirmed in this study.

THE SENTENCE WAS MODIFIED

-Lines 388-391: same as above. Change language to indicated predicted single transcriptional unit and predicted functions of proteins

THE SENTENCE WAS MODIFIED

The details of bacterial growth conditions need to be explained further. Lines 83-85 discuss only liquid culture. Did the authors never grow the bacteria on agar plates? Further, what atmosphere was used for liquid growth? at higher ODs reported I would expect a microaerophilic atmosphere to be needed, or the addition of catalase/oxyrase to the liquid media to prevent bacterial death.

MORE DETAILS WERE ADDED

Lines 44-46 do not support the current understanding of the pneumococcal pangenome.  Only 1/3 of the genome is typically shared between strains, not 70-80%, and the majority of variation is in the accessory genes expressed, not allelic variation in core genes.

THE INTRODUCTION WAS AMENDED INTRODUCING DATA FROM VAN TONDER ET AL, 2019 DOI: 10.3389/FMICB.2019.00317 AND THE REFERENCE WAS ADDED TO THE REFERENCE LIST

Lines 219-221 do not make sense, if about half the genes are on the sense and the antisense strands, what is meant by "the majority are transcribed in the same direction as DNA replication"?

WE CHANGED THE SENTENCE AS SUGGESTED

Lines 332-334: Provide a description for the virulence functions of YopX and E family proteins in pneumococci

THE SENTENCE WAS MODIFIED WITH THE DESCRIPTION SUGGESTED AND REFERENCE VIBOUD ET AL., 2005 DOI: doi:10.1146/annurev.micro.59.030804.121320. WAS ADDED TO THE REFERENCE LIST

Lines 379-380 do not make sense, how does an insertion produce a deletion?

DELETION IS PROBABLY A CONSEQUENCE OF THE INSERTION OF NOVEL GENETIC MATERIAL, SIMILARLY TO WHAT HAPPENS DURING RECOMBINATION WITH ALLELIC REPLACEMENT

Lines 487-488: this hypothesis should also be introduced earlier in the paper, this provides an interesting scientific rationale for the study, which this reviewer did not appreciate until the end of the paper. More cursory readers might miss this interesting thought question instead thinking of this as a mere genome report.

THE LAST 2 SENTENCES OF THE ABSTRACT WERE REPHRASED AND TWO SENTENCES WERE ADDED AT THE END OF INTRODUCTION

Line 536-537: please include this unpublished data.

DATA ON TRASFORMATION FREQUENCY WERE ADDED AS REQUESTED

Comments on the Quality of English Language

Figure 4 has a misspelling of lysogeny

THE TERM WAS CORRECTED

Reviewer 2 Report

The authors successfully determined the complete genome sequence of the Streptococcus pneumoniae type 19F clinical isolate BM6001, which served as the original host for the ICE Tn5253. This accomplishment enabled the identification of the mobilome, which encompasses not only the aforementioned Tn, but also two novel elements: IME Tn7089 and transposon Tn7090. In addition, the researchers discovered three prophages, two satellite prophages, five genomic islands, several ISs, RUPs, BOX elements, and SPRITEs.

Overall, this manuscript exhibits strong writing, a well-designed methodology, a clear presentation of findings, and appropriate use of references.

Consequently, I am of the opinion that this manuscript is suitable for publication in its current form.

Author Response

Thank you for the accurate review and understanding of our manuscript